# *Phytophthora palmivora*–Cocoa Interaction

**DOI:** 10.3390/jof6030167

**Published:** 2020-09-09

**Authors:** Francine Perrine-Walker

**Affiliations:** 1School of Life and Environmental Sciences, The University of Sydney, LEES Building (F22), Camperdown, NSW 2006, Australia; marie.perrine-walker@sydney.edu.au; 2The University of Sydney Institute of Agriculture, 1 Central Avenue, Australian Technology Park, Eveleigh, NSW 2015, Australia

**Keywords:** black pod rot, Oomycota, *Theobroma cacao* L., infection, stem canker

## Abstract

*Phytophthora palmivora* (Butler) is an hemibiotrophic oomycete capable of infecting over 200 plant species including one of the most economically important crops, *Theobroma cacao* L. commonly known as cocoa. It infects many parts of the cocoa plant including the pods, causing black pod rot disease. This review will focus on *P. palmivora*’s ability to infect a plant host to cause disease. We highlight some current findings in other *Phytophthora* sp. plant model systems demonstrating how the germ tube, the appressorium and the haustorium enable the plant pathogen to penetrate a plant cell and how they contribute to the disease development in planta. This review explores the molecular exchange between the oomycete and the plant host, and the role of plant immunity during the development of such structures, to understand the infection of cocoa pods by *P. palmivora* isolates from Papua New Guinea.

## 1. Introduction

Within the order Peronosporales, the largest genus with over 120 described species, *Phytophthora* is a hemibiotrophic phytogen capable of infecting a wide range of hosts, including many agricultural crops, worldwide [1,2]. One of the most economically important and delicious crops affected is cocoa (*Theobroma cacao* L.). At a global scale, black pod or pod rot is the most important cocoa disease, caused by several *Phytophthora* species (Table 1) and contributing to significant pod losses of up to 30% and killing up to 10% of trees annually [1,3,4]. Some black pod-causing *Phytophthora* species have distinct geographical distributions (Table 1) while *Phytophthora palmivora* (Ppal, Butler) [5], which was originally isolated from Palmyra palm (*Borassus flabellifer*) in 1907, has a pantropical geographical distribution and is found in virtually all cocoa production areas [1,4,6]. In addition, it has a wide host range of over 200 plant species in the tropics [1,4,6]. This has serious implications for smallholder farmers who produce over 80% of all cocoa, as cocoa trees are mainly grown under shade trees, either in an inter-cropped or in semi-natural agro-forestry systems [3]. 

## 2. Morphology of *P. palmivora* (Ppal) 

*Phytophthora*, as an oomycete, is part of a distinct group of fungus-like eukaryotic microbes. It shares a range of morphological features with fungi, but it possesses other features unique to plants, such as the major component of its cell wall being cellulose, unlike true fungi, which consists mainly of chitin [11]. Another feature is that its mycelium is composed of hyaline, branched, non-septate filaments, while fungal hyphae have septate. 

The dispersal of *Phytophthora* by wind or water is achieved by asexual sporangia (Figure 1A), which develop at the ends of specialized hyphal tips [12]. Sporangia morphology can be quite diverse but the shapes of Ppal sporangia range from ovoid-ellipsoid to obpyriform, and they are papillate and cadacous, i.e., short pedicels [1,12,13]. Sporangia can germinate directly forming germ tubes and hyphae, or they release motile asexual spores called zoospores (Figure 1D–G). Anodotactic *Ppal* zoospores actively swim with the aid of two flagella on the wet surface of plant tissues or in flooded soil by negative geotaxis [14], by electrotaxis in natural root-generated electric fields [15] and by chemotaxis [16,17,18]. In addition, high humidity/moisture and splashes of water help in the spread of such zoospores from plant to plant. Therefore, the reduction of high humidity and avoidance of excess water are some of the practices in greenhouses/glasshouses for *Phytophthora* disease control [19].

Ppal reproduces both asexually, via the motile zoospores, and sexually, via the formation of oospores caused by the contact of two structures found at the mycelium tips: the female oogonium (the sac which contains the developing oospore) and the male structure, the antheridium. *Phytophthora* species can be described as homothallic (self-fertile) or heterothallic (self-sterile) where the latter requires the mating of compatible A1 and A2 types. Ppal is heterothallic and oospores can be produced only when A1 and A2 types are grown together on agar plates or on infected plants. Interestingly, in Ppal, the A2 compatibility type is predominant on cocoa throughout the world [20,21]. 

Chlamydospores are usually globose and can be intercalarily or terminally located on the mycelium. They can be distinguished from hyphal swellings due to the presence of septate (Figure 1B–D). They are recognized as resistant, long-term survival structures [22]. It was shown that Ppal storage cultures can remain viable in water at room temperature for up to 23 years and that Ppal colonies developed from chlamydospore-like structures that were produced in the absence of adequate nutrition and aeration [23].

## 3. *P. palmivora* (Ppal)’s Infection Process in Cocoa

Ppal belongs to Clade 4, whose species form papillate sporangia and are known to be pathogenic to plant roots [7], causing root rot disease in many plants [24]. It can infect other plants tissues such as the stems, leaves and fruits of many economically important tropical plants such as breadfruit (*Artocarpus altilis*), coconut (*Cocos nucifera*) and durian (*Durio zibethinus*) [1,23], including both monocots and dicots. Infection studies have been done in model plants such as *Medicago truncatula* [25,26], *Nicotiana benthamiana* [27,28] and in the model liverwort, *Marchantia polymorpha* [29], as well as on coconut [30], oil palm [31,32,33], betelvine [34], citrus hosts [35,36], rubber [37], and papaya [38]. 

In cocoa, Purwantara [39] demonstrated that soil from cocoa plantations in West Java was a massive and consistent source of Ppal inocula and that Ppal infection from soil to the cocoa pods appears to be mainly through contact or rain splash. Caducous sporangia or motile zoospores adhere to the plant surface. Though a single sporangium could germinate to start the infection cycle within cocoa, our focus will be on the infectious agent, a single motile zoospore. The zoospore adheres to the plant surface where it sheds its flagella and forms a non-motile spherical cyst (Figure 1H–I and Figure 2B).

Encystment and cyst germination are two important developmental stages required for Ppal to adhere or to dock [40,41] on the surface of cocoa plant tissues. Studies in various oomycetes have demonstrated that the zoospores docked precisely on the root surface at its ventral face with the help of the posterior flagellum, allowing the deposition of adhesive contents during encystment on the plant host and orientating *Phytophthora* to germinate toward the host [42,43,44]. Transient leaching treatments of encysting zoospores, which involved leaching solutions at various time intervals underneath polycarbonate membranes, calcium, pectin and various other molecules as well as mechanical agitation, affect the ability of Ppal to dock and to form germ tubes [45,46,47]. Further work by Zhang and colleagues [48] found that methylation destroyed the capacity of the pectin to induce germination, but its methylated form induced zoospore rounding and partial encystment at low concentrations. This is important as the outer surface of most aerial organs of plants such as leaves, flowers, fruits and non-woody stems are covered with the cuticle, which consists of cellulose, hemicelluloses and pectins [49]. Using functional and structural analyses, pectin methylesterase-coding genes have been found in various *Phytophthora* species [50,51,52] as well as polygalacturonase and pectate lyase in *P. capsici* [53], capable of degrading pectin. 

Bimpong and Clerk [16] demonstrated that Ppal zoospores responded chemotactically to an extract of cocoa pod where the cyst germ tubes grew towards the stimulus but not to the exudate. The germ tube grows on the plant surface (Figure 2C) and various environmental cues induce the formation of an appressorium for the subsequent entry into the plant host cell [54]. Studies by Ali et al. [55] and Tey [56] demonstrated that Ppal cysts formed germ tubes and appressoria on cocoa pod husks and leaf tissues. Entry via wounding and stomatal pores has been observed by Ppal in cocoa and in other *Phytophthora*–plant infection studies by microscopy. Studies in chickpea showed that *P. megasperma* f. sp. *medicaginis* vacuolated zoospore cysts formed germ tubes to gain entry into stomatal pore and after septum formation, differentiated into the primary hypha within the hypocotyl region [57]. Widmer et al. [36] found that *P. palmivora* gained entry in a natural wound site on a root of tolerant trifoliate orange (*Poncirus trifoliata*) via a germ tube. 

There are differences in the structure and organization of various plant tissues; for example, the presence of the cuticle on the aerial parts of plants [49] i.e., the leaf, the non-woody stem and pod, but not in the roots. Figure 2 is a schematic diagram focusing on the general infection process of *Phytophthora* in plant tissues (Figure 2D–F). The appressorium forms a penetration peg to penetrate the cuticle layer or the cell wall of an epidermal cell [58,59,60]. In ground tissues, germ tubes that emerged from cysts penetrated the root epidermis, usually by intercellular growth along the anticlinal cell walls [61] or by appressorium-mediated penetration via a penetration peg between two rhizodermis cells by *P. parasitica* [62]. Intracellular penetration can occur and germ tubes from encysted zoospores can become swollen and produce a penetration peg [57]. Then, specialised hyphae invade plant cells to form haustoria (Figure 2D–E) [26,63,64,65,66,67]. Histological studies in *Quercus ilex* roots during *P. cinnamomi* infection found haustoria-like structures in the cortical root and phloem cells [63]. Haustoria have been observed in *Medicago* root epidermal cells [64]. The Ppal haustorium is a short, swollen, anucleate hyphal branch, which protrudes into the peripheral cytoplasm of the host cell [66,68]. The haustorium is surrounded by a specialized host-derived membrane, the extrahaustorial membrane (EHM), which is distinct from the plant plasma membrane. In fungi, haustoria function as feeding structures [69]. During this phase of growth, Ppal interaction with its host is biotrophic and secreted effectors and enzymes targeting the apoplastic and cytoplasmic sites in the plant host have been shown to play a role in plant cellular reprogramming/rearrangement and in reducing plant immunity [65,70,71,72,73,74,75,76,77,78,79,80]. Therefore, the development of the haustorium plays a critical role in the successful parasitic infection of *Phytophthora*.

Intercellular infection by Ppal can be observed *in planta* as well [33,81]. To complete its lifecycle, the hemibiotrophic Ppal switches from a biotrophic to a necrotrophic lifestyle highlighted by the presence of necrotic plant tissues, prolific hyphal growth and the formation of sporangia as well as chlamydospores in plant tissues (Figure 2F) [28,29,60,82].

## 4. Overcoming Plant Host Immunity by Ppal and Other Oomycetes

*Phytophthora*, along with other plant pathogens, needs to overcome the plant host’s immunity. In the first line of defense, the cocoa plant would use pattern-recognition receptors (PRRs) found on the plant cell membrane. These detect microbe- and pathogen-associated molecular pattern (MAMP and PAMP) molecules leading to pattern-triggered immunity (PTI). In addition, the cocoa plant needs a secondary line of defense, as *Phytophthora* can overcome PTI by secreting effectors that suppress PTI responses, resulting in effector-triggered susceptibility. These effectors can act within the apoplastic and symplastic regions of the plant cell, where a secretory system would enable the delivery of such effectors via the appressorium and the haustorium respectively [59,83]. Plants possess cytoplasmic resistance (R) proteins that recognize such effectors. These R proteins are intracellular receptor proteins of the nucleotide binding–leucine-rich repeat (NB-LRR) type [84,85,86], which are activated in the presence of key effectors to trigger a hypersensitive response (HR) and systemic acquired resistance (SAR) in the plant host [87,88,89]. This is termed effector-triggered immunity (ETI). It is pathogen strain- or race-specific and associated with programmed cell death [90]. Furthermore, Thomma et al. [90] proposed that PAMP receptors and R proteins are part of the plant’s surveillance mechanism and that both PTI and ETI are used for effective immunity. 

Under a hypersensitive defense response, a rapid plant cell death occurs at the point of pathogen ingress and is generally associated with ETI. Recent work by Gu et al. [91] observed the upregulation of multiple plant NB-LRR genes in Mexican wild potato species, *Solanum pinnatisectum* against *P. infestans,* where hyphal expansion was significantly restricted in epidermal cells and mesophyll cell death was predominant at 12 hours post inoculation (hpi), thus indicating that the HR was induced upon infection. Under SAR, a localised response due to a pathogen induces resistance at sites remotely located from the initial infection, and this is associated with the transport of defense signals such as salicylic acid throughout the plant, resulting in broad-spectrum disease resistance against secondary infections. Recent work in potato has shown a link between microRNAs i.e., non-coding RNAs that act as negative regulators of gene expression, in SAR response [92]. The knockdown (KD) of miR160 compromised SAR response to *P. infestans* in miR160 KD lines of *S. tuberosum* cv. Désirée [92]. miRNAs also affected NB-LRR genes in tomatoes [93] and in soybean [94] during *P. infestans* and *P. sojae* infection respectively. In addition, it has been shown that *P. sojae* secreted effectors to suppress RNA silencing in plants by inhibiting the biogenesis of small RNAs [95], thus promoting infection. Recent evidence suggests that miRNAs repression of NB-LRR resistance genes in plants is not only used by plant pathogenic oomycetes such as *Phytophthora,* but could play a role in the infection of leguminous plants by symbiotic bacteria. miRNAs repressed NB-LRR resistance genes to promote *Sinorhizobium meliloti*’s colonization and the development of nitrogen-fixing nodules in *Medicago truncatula* [96]. Sós-Hegedűs et al. [96] proposed a model that a subset of NB-LRR-targeting plant miRNAs (miR482/2118 superfamily, miR1507, miR2109) could tip the balance in NB-LRR proteins in the *M. truncatula*, affecting the perception of *S. meliloti* as a pathogen or a symbiont [96]. 

Hardham and Blackman [97] and Wang and Jiao [98] highlighted PAMPs and effectors used in other characterized plant pathogenic *Phytophthora* species such as *P. infestans*, *P. capsici*, *P. cinnamomic*, and *P. parasitica* and some of the approaches used to understand their role in PTI and ETI. Furthermore, Raaymakers and Van den Ackerveken [99] listed several oomycete-derived patterns known to activate plant immunity. In the case of Ppal and cocoa interaction, Ppal success in establishing disease would rely on avoiding detection of PAMPs by PRRs or the secretion of effectors within the plant’s apoplast and symplast to interfere with PTI or ETI to support its infection and promote disease development. The following section of this review will focus on some specific Ppal-derived patterns such as lectins and Ppal RxLR effectors and their functions during infection in cocoa and in other model plants.

### 4.1. Necrosis and Ethylene-Inducing Peptide 1 (Nep1)-Like Proteins

Necrosis and ethylene-inducing peptide 1 (Nep1)-like proteins (NLPs), which were first identified in *P. parasitica*, have been shown to induce necrosis in planta [100,101]. Work by Schumacher et al. [102] identified NLPs in the obligate biotrophic oomycete *Plasmopara viticola,* which causes grapevine downy mildew. In addition, NLPs are secreted by bacteria and fungi and come in two forms, those that are cytotoxic to eudicot plants and those that are noncytotoxic [103]. Within 24 h of application of Nep1 purified from *Fusarium oxysporum* f. sp. *erythroxyli* culture filtrates (at 5 μg ml^−1^ plus 0.2% Silwet-L77), the majority of stomata guard cells and two or more neighboring epidermal cells around each affected stomata on the abaxial leaf surface in mature green cocoa leaves were killed, with the microscopic necrotic flecks and darkly pigmented necrotic lesions developed on Nep1-treated field-grown Amelonado cocoa pods (at the same concentration) [104]. It was suggested that lesion development in cocoa pods was due to Nep1 entry via the stomata on the cocoa pod surface [104]. In addition, the expression of cocoa genes involved in defense gene regulation, cell wall development and energy production were different in young red leaves and mature green leaves of cocoa in response to the application of Nep1 [104]. Bae et al. [105] demonstrated that six of the nine NEP1 orthologues, which had a similar sequence to the NEP1 of *F. oxysporium*, were expressed in *P. megakarya* mycelium and in *P. megakarya* zoospore-infected cocoa leaf tissue using leaf disc assays. Evangelisti et al. [28] identified 24 putative NLPs in the Ppal secretome study in *N. benthamiana*. Ali and colleagues [106] identified several NPP1-type necrosis inducing-like proteins and NPP1-like proteins, a necrosis-inducing protein NPP8 and a Suppressor of Necrosis 1 protein (SNE1) in Ppal–cocoa infection studies. The latter, *SNE1*, previously characterized in *P. infestans*, was shown to translocate to the plant nucleus and suppressed the action of secreted NLPs from *Phytophthora* that are expressed during the necrotrophic growth phase, as well as programmed cell death mediated by the Avr3a/R3a protein interaction [107].

### 4.2. Lectins and Cellulose-Binding Elicitor Lectins (CBELs)

Plant lectins play a signaling role to modulate plant immunity responses to various plant pathogens via lectin receptor kinases [108,109]. Previous work in potatoes demonstrated that lectins lysed *P. infestans* zoospores and mediated the binding of cell membranes of potato to cell wall surfaces of infecting hyphae of both compatible and incompatible races of *P. infestans* in vivo [110,111]. By expressing the Arabidopsis lectin receptor kinase LecRK-I.9 gene in potato and *N. benthamiana*, late blight resistance to *P. infestans* was significantly enhanced [112]. Cellulose-binding elicitor lectins (CBELs) are cell wall-localized glycoproteins involved in cell wall organization and the adhesion of the mycelium to cellulosic substrates [113,114,115]. They have also been shown to aid in *Phytophthora*’s penetration into its plant host by mediating the oomycete’s attachment to the host surface [113,114,115]. According to Khatib and colleagues [116], this glycoprotein is widespread in the genus *Phytophthora*. Secretome work on Ppal identified 24 lectins including one CBEL [28]. Infection work in Ppal on cocoa found eight CBELs and a putative CBEL-like protein transcribed in the Ppal mycelia, zoospores and in planta [106]. Work by Laroque and colleagues [117] found that CBEL played a role in triggering immunity in the *P. parasitica–Arabidopsis* interaction, showing that BRASSINOSTEROID INSENSITIVE 1-associated kinase 1 (BAK1) and NADPH oxidase genes were required for CBEL-induced oxidative burst and defense responses but not for necrosis.

### 4.3. Elicitins

Lack of extracellular 10-kDa elicitins have been correlated with virulence in most *P. parasitica* isolates of tobacco [118]. Work by Huitema et al. [119] identified two classes of elicitins that are secreted such as INF1 (class I) and the cell-surface-anchored polypeptides, INF2A and INF2B (Class III) in P. *infestans*. Coexpression of INF1 and the NLP protein PiNPP1.1 from *P. infestans* led to synergistic enhancement of cell-death elicitation in *N. benthamiana* [120]. Work by Le Fevre and colleagues [60] demonstrated that PAL1, the Ppal homolog of *P. infestans inf1*, was transcriptionally induced in barley roots and leaves during Ppal infection. Ppal produces a 10-kDa protein, palmivorein [121,122] and a 75 kDa elicitor, which triggered defense responses in rubber plants [122]. In another study, a crude elicitor from culture filtrates of Ppal was applied to rubber tree leaves and this pretreatment significantly increased Ppal infection in such leaves [123]. In addition, infiltration of this crude elicitor promoted cell death and increased salicylic acid (SA), abscisic acid (ABA) and the phytoalexin, scopoletin (Scp) content in tobacco and rubber tree leaves [123]. Recent work by Pettongkhao and colleagues [124] isolated a secreted glycoprotein of 15 kDa from a papaya Ppal isolate and suggested that Ppal15kDa played an important role in normal development of Ppal infection structures. All Ppal15kDa mutants generated via CRISPR/Cas9-mediated gene editing, were compromised in infectivity on *N. benthamiana* and papaya [124]. In addition, the mutants’ development was also affected as they produced smaller sporangia, shorter germ tubes, and fewer appressoria, leading to reduced levels of pathogenicity [124].

### 4.4. Glycoside Hydrolase 12 Proteins

Ma et al. [125] showed that the *P. sojae* glycoside hydrolase 12 protein, PsXEG1, acted as a PAMP in soybean (*Glycine max*) and solanaceous species and, by both silencing and overexpression of XEG1 in *P. sojae*, severely reduced virulence. Later, Ma et al. [126] demonstrated that *P. sojae* secreted a paralogous PsXEG1-like protein, PsXLP1, that had lost enzyme activity. The latter could bind to a soybean apoplastic glucanase inhibitor protein, GmGIP1, more tightly than did PsXEG1, thus freeing PsXEG1 to assist *P. sojae* infection [126]. *P. parasitica* orthologs PpXEG1 and PpXLP1 were found to have similar functions and both genes were found to be conserved in other *Phytophthora* species [126]. Use of the Carbohydrate-Active EnZymes (CAZy) database enabled Zerillo et al. [127] to identify xyloglucan-β-1, 4-D-endoglucanase genes in family GH12 in *Pythium* sp. and various oomycetes. Evangelisti et al. [28] identified putative glycosyl hydrolases in Ppal in *N. benthamiana* infection studies. Only nine belonged to GH12 family, where PLTG_13824/PEX_0219 was described as a cell 12A endoglucanase and the remaining eight as hypothetical proteins [28]. However, work by Ali et al. [106] identified two candidate genes in Ppal and three candidate genes in *P. megakyara* belonging to the glycoside hydrolase 12 family.

However, recent work by Ochola and colleagues [128] may provide some clues as to how *Avr* gene expression impacts the compatibility of plant disease. By using the CRISPR/Cas9 engineering technique, *PsAvr3b* promoter sequences from *P. sojae* were substituted in situ with promoter sequences from Actin (constitutive expression), PsXEG1 (early expression), and PsNLP1 (later expression). Compared to the wild type and the unedited mutant (T1) i.e., with the native *PsAvr3b* promoter as controls, *PsAvr3b* expression was significantly reduced when the *PsAvr3b* promoter was substituted with PsXEG1 (early expression) or PsNLP1 (late expression) promoters [128]. When these promoter mutants carrying PsXEG1 (X02 and X03) or PsNLP1 (N02 and N10) were tested on Williams (susceptible) and two resistant (*Rps3b* and *Rps3c*) soybean cultivars, these mutants gained virulence against the resistant *Rps3b* cultivar while mutants containing the PsACT promoter (A24 and A26) were unable to infect soybean cultivars carrying *Rps3b* [128]. No infection was observed with the WT and T1 control on soybean cultivars carrying *Rps3b* [128]. Further transcriptomic studies with these promoter mutants highlighted a difference in gene expression in the resistant *Rps3b* cultivar such as the wound-inducible, jasmonate synthesis-degradation lipoxygenase (LOX-1) and the proline extensin-like receptor kinase 1 (PERK1) [128]. Compared to the WT strain, LOX-1 and PERK1 were upregulated in soybean cultivars (carrying *Rps3b*) infected with the mutant with the PsACT promoter (A24) expressed, while they were downregulated in those infected with the promoter mutants, PsXEG1 (X03) and PsNLP1 (N02), respectively, at 24 hpi [128]. 

### 4.5. Transglutaminases (Pep-13)

The calcium-dependent cell wall transglutaminase (TGase), GP42 from *P. sojae,* consists of a peptide fragment/domain (Pep-13), which activates plant defense in parsley and potato [129]. GP42 belongs to a group of enzymes that catalyzes the post-translational modification of proteins by the formation of isopeptide bonds [130]. In a proteome study of *P. infestans* membrane, two transglutaminases were encoded by PITG_22117 and PITG_16956, respectively [131]. PITG_22117 was detected in both non-sporulating mycelium and germinating cysts with appressoria while PITG_16956 was identified from sporulating mycelium [131]. Potato plants treated with Pep-13 not only were able to mount a salicylic acid (SA)- and jasmonic acid (JA)-dependent defense response, but were also found to activate the co-receptor BAK1 [132]. Recent work by Wang et al. [133] demonstrated that the *Phytophthora* MAMP Pep-13 triggered SOMATIC EMBROYOGENESIS KINASE 3 (SERK3)/BAK1-independent PTI. In wild potato (*Solanum microdontum*), a receptor-like protein ELR (elicitin response) mediated extracellular recognition of the elicitin domain, a domain known to be conserved in *Phytophthora* species. ELR also was associated with the immune co-receptor BAK1/SERK3 and the transfer of ELR into cultivated potato resulted in enhanced resistance to *P. infestans* [134]. Previous work by Brunner et al. [129] found a GP42-like protein containing the Pep-13 motif in Ppal and in the Ppal secretome study; Evangelisti et al. [28] identified five out of six transglutaminases carrying the conserved Pep-13 motif.

### 4.6. RxLR and CRN Effectors

Many effectors are known to act in the apoplastic and symplastic region of plant cells during the appressorium and the haustorium development [59,83]. Two classes of effectors are known: RxLR where N terminus of such effectors have a conserved arginine-any amino acid-leucine-arginine motifs usually linked with a glutamic acid-glutamic acid-arginine domain (RxLR-dEER). The other class is CRinkling- and Necrosis-inducing proteins (CRNs), which contain a LFLAK motif. These are involved in manipulating many functions linked to the host immunity such as cell protease function, phytohormone signaling and RNA silencing effectors [97,98]. Secretome studies in Ppal identified putative secreted proteins such as RXLR effectors [28]. Transcriptomic work found four RxLR effectors (REX1-4) to be upregulated during Ppal infection in *N. benthamiana* roots, and REX2 and REX3 effectors were found to suppress host secretion [28]. In *P. parasitica*, the Penetration-Specific Effector 1 (PSE1) protein is a secreted RxLR effector protein whose expression is induced during appressorium-mediated penetration of the host roots, but declines during early biotrophy and cannot be detected during the necrotrophic phase of infection [59,135]. PSE1 abolished cell death in tobacco plants triggered by the *P. cryptogea* elicitin cryptogein and the *Pseudomonas syringae* AvrPto avirulence protein and increased susceptibility of *A. thaliana* to *P. parasitica* by altering the distribution of key auxin efflux transporters in PSE1 transgenic *A. thaliana* lines [135]. Genome, transcriptome and secretome studies combined with RNA-sequencing and RT-PCR identified RXLR effectors and crinklers in Ppal and *P. megakarya,* which were differentially expressed in mycelia, zoospores, and in planta (infected pod husks) [106]. Furthermore, recent work by Morales–Cruz et al. [136] predicted that Ppal had 717 RxLR effectors compared to *P. megakarya,* which had 1,382 effectors due to genome duplication and expansion in the latter. In addition, 251 “putative effectors” in Ppal had shared homology and often bordered RxLRs [136]. More work would be needed to understand the functions of these effectors in Ppal and how they aid in infection and in manipulating cocoa’s immunity to cause disease.

## 5. Cocoa Diseases by *P. palmivora* (Ppal)

*T. cacao* is the only species within the *Theobroma* genus that is cultivated by about 6 million farmers globally [137]. The species is divided into three main recognized genetic groups: Criollo, Forastero and Trinitario [138,139]. The latter is a hybrid from crosses between the Criollo and Forastero varieties and is cultivated in many parts of the world due to its aromatic, high-yielding and disease-resistant characteristics [137,139]. Some cocoa breeding programs have been focused on selecting lines resistant to many plant pathogens as well as Ppal [140]. 

Ppal causes two main types of disease on cocoa trees: black pod and stem canker. Figure 3 shows a mature healthy cocoa plant growing in a glasshouse, highlighting the target sites of Ppal infection. In black pod, pods or cherelles (immature pods, Figure 3E,F) can be infected at any place on the surface, however, initial infection is usually at the tip or stem end. 

Studies in 12 diverse cocoa genotypes demonstrated that germinating zoospores of Ppal could penetrate through stomata, epidermal hair base, scar and by direct penetration of pods [141,142]. Symptoms are a brown or black spot on the pod, which spreads to cover the whole pod.

In stem canker, Ppal mycelia spread from infected pods [143] along the stalk into the flower cushions (Figure 3B) and further along the stem or via direct infection in wounds along the stem. Newly infected bark may not show any external symptom, but the cambial layer would be infected [144]. Symptoms of canker are the formation of reddish water-soaked lesions with dark brown to black margins, and in some cases, reddish-brown liquid oozed from these lesions, usually through cracks in the bark [143,144]. In Sulawesi, incidence and severity of stem canker in cocoa by Ppal increases during the wet season, especially in more susceptible genotypes [145]. Okey and colleagues [146] demonstrated a strong correlation between bark hardness and moisture content with canker resistance to Ppal in greenhouse studies. The same authors [146] proposed that extra-xylary tissue hardness associated with fiber content or deposition of suberin, callose and lignin could hinder the progress of fungal pathogens and that bark hardness, acting as a mechanical barrier, had contributed to the slow rate of tissue colonization of Ppal in the canker-resistant cocoa line used in greenhouse studies, leading to the use and selection of resistant cultivars with acceptable horticultural traits [147]. Such traits in Ppal resistant lines were related to lignin concentration in cocoa stems [148] and the high activities of plant enzymes such as peroxidase (PO) and polyphenoloxidase (PPO), which are involved in phenol oxidation and lignin production, and phenylalanine ammonia-lyase (PAL) in lignin and phenol biosynthesis in response to Ppal infection in cocoa-resistant clones [149]. In another model system, it has been shown that both peroxidase activity and lignin deposition increased in the cell suspensions of the resistant *Capsicum annuum* (pepper) variety to *P. capsici* elicitors compared to the susceptible or intermediate pepper varieties [150].

Other approaches to controlling stem canker in cocoa have involved the application by trunk injection of potassium phosphonate (phosphite) [151]. Potassium phosphonate has been used as a systemically translocated chemical to protect plants against oomycetes due to its ability to induce rapid and localized defense responses similarly observed in phosphonate-treated *A. thaliana* seedlings inoculated with Ppal zoospores [67]. 

Other parts of the cocoa plants can be infected by Ppal i.e., the flower cushion, the chupons, leaves (Figure 3A–F) and seedlings, as well as the roots [20,82,152,153,154]. Work on cocoa roots by Oppoku and Wheeler [155] demonstrated that Ppal persisted in association with roots for at least 6 months, and the recovery of the oomycete generally declined with time.

However, cocoa as a perennial plant takes a long time to grow and the selection of Ppal-resistant lines or germplasm requires quick, easy and cheap inoculation testing methods [141,142]. Detached or attached cocoa leaves and pods are some of the materials used to determine resistance [141,142]. Work by Iwaro and colleagues [142] tested leaves and pods of various clones for resistance and demonstrated that there were two levels of resistance in both organs. Their studies showed a poor relationship between pod and leaf reaction at Ppal penetration stage of infection while a high positive correlation was observed between pod and leaf resistance at the post-penetration stage of infection, suggesting the role of a systemic mechanism in post-penetration resistance [141,142]. Resistance can be effective from the point of entry of the pathogen (penetration) or at a later stage during its development within the host tissue (post-penetration) [156]; thus, penetration and post-penetration resistance can both be used as selection criteria in breeding to improve the existing levels of cocoa resistance to Ppal. In expression pattern studies in susceptible cocoa pods, Ali and colleagues [9,55] highlighted the differences between Ppal and *P. megakarya*, especially in Ppal-inoculated wounded pod pieces, where Ppal is known for its rapid progression when penetrating through wounds. Previous studies in betelvine and papaya have shown that there is a synergistic effect with plant pathogenic nematodes, *Rotylenchulus reniformis* and *Meloidogyne incognita,* which predispose these plants to attack by Ppal [157,158,159]. 

## 6. Ppal Isolates from Papua New Guinea Cocoa Plantations

Analyses of random amplified microsatellites (RAMs) of 263 of the *Phytophthora* isolates demonstrated that there was limited morphological, physiological and genetic diversity of Ppal isolates from cocoa pods in Papua New Guinea (PNG), and that Ppal from cocoa in PNG formed a single, continuous largely asexual population [13,160] (Figure 1A–N). Recent studies in the genetic diversity among 81 Ppal isolates from various host plants and geographical regions in Indonesia and Japan using rep-PCR (BOX, ERIC, REP and M13) and microsatellite markers demonstrated that the isolates clustered into six groups, which corresponded more to geographic regions rather than host plants or mating types [161]. These studies highlighted the importance of implementing key quarantine measures to prevent the spread of Ppal-contaminated plant materials to different geographical regions [160,161]. 

However, work by Appiah et al. [162] demonstrated that Ppal isolates from different geographical sources associated with black pod disease in cocoa showed considerable inter- and intra-specific morphometric variation. This is important as correct identification of the pathogen is crucial, since Ppal can be controlled by crop sanitation alone, whereas *Phytophthora megakarya* (Table 1) cannot [9,163,164]. Through sexual reproduction or interspecific hybridization, *Phytophthora* could gain allelic diversity and achieve large sexual/clonal population sizes through rapid proliferation [165]. These would enhance pathogen fitness by generating recombinant genotypes that may be more pathogenic or resistant to crop protection chemicals [166].

## 7. Conclusions

The question remains as to why these Ppal isolates formed clusters based on their geographic regions and what are the characteristics that have allowed such isolates to infect and be pathogenic to current cocoa lines. Goodwin [167] presented many factors that could contribute to the genetic variation in *Phytophthora* population. Migration of *Phytophthora*, via the introduction of contaminated plant materials to different geographical regions or from centers of origins, would put pressure on founder *Phytophthora* populations [167]. Such populations would be subjected to genetic drift due to changes in environmental conditions; selection would contribute to overall fitness and, mating as heterothallic species, should contain high level of heterozygosity [167]. Brasier [168] proposed that the soil, with *Phytophthora* resting inoculum oospores and chlamydospores, would be a reservoir of genetic variation but the reinfection of the hosts would exert strong directional selection on such variation, favoring genotypes capable of infecting a particular host species or part of the host. Furthermore, successful pathogen genotypes could be maintained by asexual reproduction by directional and stabilizing selection as long as the host is still available [168]. Under episodic selection during widespread and continuous crop monoculture or following the introduction of a new and susceptible host population for example, rapid speciation could occur, increasing specialization on a single host species [168,169]. Combined with asexual reproduction and pathogenic feedback, this would lead to a reduction in genetic variability and to the emergence of a clone [169]. 

In the case of the characterized PNG cocoa Ppal isolates [160], a study investigating the differences in gene expression related to PTI and ETI during cocoa pod and stem infections would be useful in understanding the differences in pathogenicity observed in cocoa plantation fields at different locations in PNG.

## Figures and Tables

**Figure 1 jof-06-00167-f001:**
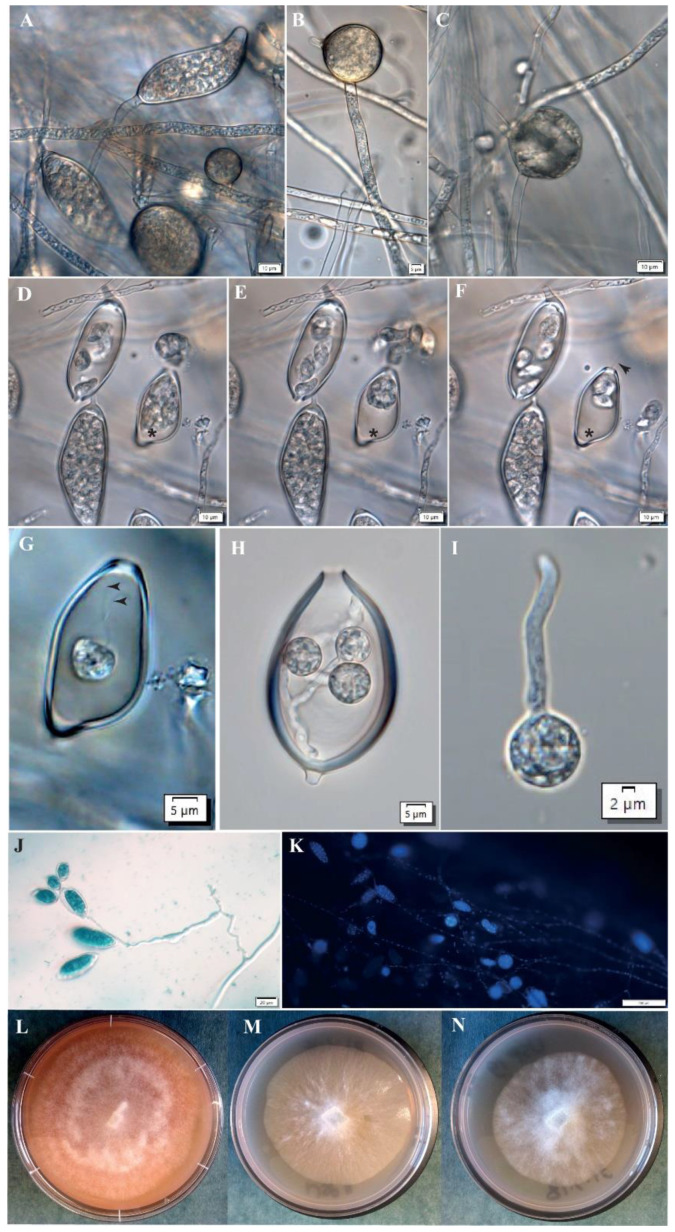
The characteristic morphology of *Phytophthora palmivora*. These Ppal isolates MAG14, NSP11 and NSP19 were from infected cocoa pods from three different farms in Madang and Bougainville (PNG) in 2005 [13]. NSP11 and NSP19 were from two different districts, Sinai and Buin, respectively in Bougainville. (**A**) Papillate sporangium (mature); (**B**) terminal chlamydospore forming new hyphal extension; (**C**) intercalary chlamydospore; (**D**–**F**) release of zoospores within 1 min time period in one sporangium (black asterisks); (**G**) trapped zoospores; (**H**) trapped cysts germinating within a sporangium; (**I**) germinating cyst on a glass microscope slide; (**J**) sympodial branching of sporangiophore with papillate sporangia stained with lactophenol blue; (**K**) DAPI (4′,6-diamidino-2-phenylindole) staining of nuclei in hyphae, zoospores within sporangia and chlamydospores; (**L**) to (**N**) Stellate/striate to radiant colony types of MAG14 (l), NSP11 (**M**) and NSP19 (**N**) isolates on carrot agar post 7 d growth at 26 °C respectively. Black arrowheads highlight the presence of flagella in (**F**,**G**). (**A**,**D**–**G**,**I**–**K**) isolate MAG14 on carrot agar post 6 d growth at 26 °C; (**B**,**C**,**H**) isolate NSP11 on oatmeal agar post 12 d growth at 26 °C. Micrographs (**A**–**K**) were captured using an Olympus BX51 microscope equipped with DP74 Olympus camera under differential interference contrast (DIC) and ultraviolet (UV) fluorescence.

**Figure 2 jof-06-00167-f002:**
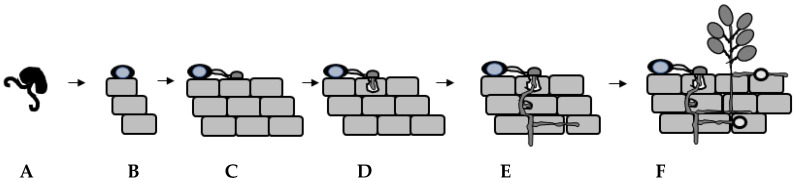
Schematic diagram of the infection process of *Phytophthora palmivora* in planta. (**A**) Via splashing of water droplets, flowing surface water, wind-driven rain, zoospores are released from the caducous sporangia and they actively swim towards potential infection sites on both aerial and subterranean surfaces of plants; (**B**) Once at the site, zoospores lose motility by shedding their flagella, encyst, and the newly formed wall adheres to the surface of the plant; (**C**) Cyst germinates forming a germ tube and later an appressorium, which provides stronger adhesion to the host surface in preparation for subsequent invasion into the epidermis of some aerial tissues. In ground tissues, the germ tube can penetrate the root epidermis by growing intercellularly along the anticlinal cell walls; (**D**) At the appressorium adhesion site, *P. palmivora* hyphae grow to invade intracellularly, forming a haustorium—this is the biotrophic stage; (**E**) During the necrotrophic stage, secondary hyphae form that kill the host cell; and (**F**) new structure characteristic of the oomycete i.e., intercalary and terminal chlamydospores and sporangia develop, providing new inocula for future infection of other regions on the same host or other hosts in the field.

**Figure 3 jof-06-00167-f003:**
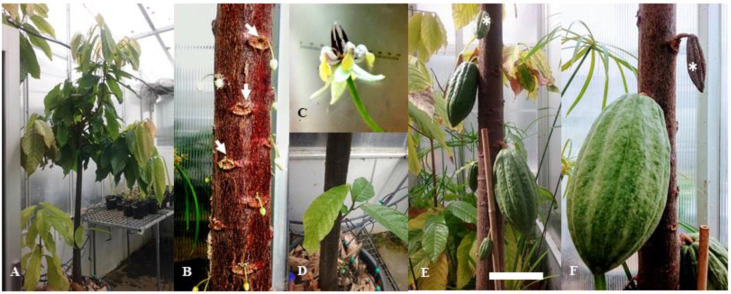
*Theobroma cacao* L. (cocoa) plants (undefined mixture of Trinitario) in the glasshouse. (**A**) Dimorphous cocoa plant; (**B**) Unopened cauliflorous flowers at different stages of growth and development attached to swollen and enlarged regions of the trunk known as flower cushions (white arrows); (**C**) Detached cauliflorous, reddish-white color, odorless and self-fertile flower; (**D**) a sucker or chupon near the base of the trunk; (**E**) cherelles (immature pods) at different stages of growth and development arising from former flower cushions and (**F**) close-up of an immature pod. White asterisk (*) in (**E**) shows the same cherelle in (**F**) after 1 month aborted in growth. Scale bar in (**E**) represents 5 cm.

**Table 1 jof-06-00167-t001:** List of *Phytophthora* species known to cause black pod in cocoa and characteristics related to their geographical distributions, host range, clade, sex, and genome size.

Species Name ^a^	Geographical Distribution ^a^	Clade ^b^	Host ^b^	Sex ^b^	Papil. ^b^	Genome Size (Mb)
*Phytophthora capsici* (Leonian)	Brazil, El Salvador, Guatemala, India, Jamaica, Mexico, Trinidad, Venezuela	2	Multiple	He	P	64.00 [7]
*P. citrophthora* (RE Smith and EH Smith)	Brazil, India, Mexico	2	Multiple	He	P	n.d
*P. heveae* (Thompson)	Malaysia	5	Multiple	Ho	P	n.d
*P. megakarya* (Brasier and Griffin)	Cameroon, Côte d’Ivoire, Fernando Po (aka Bioko), Gabon, Ghana, Nigeria, São Tomé (islands of Principe and São Tomé), and Togo	4	*T. cacao*	He	P	126.88 [8]
*P. megasperma* (Dreschler)	Venezuela	6	Multiple	Ho	NP	62 [9]
*P. nicotianae* var. *parasitica*	Cuba	1	Multiple	He	P	76.50 [2]
*P. palmivora* (Butler)	Pantropical	4	Multiple	He	P	151.23 [10]

^a^ Adapted from [6]; ^b^ Adapted from [11]; He = Heterothallic; Ho = Homothallic; n.d, not determined; Papil., papillate; NP = non-papillate sporangium; P = Papillate sporangium.

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
