# Peer review of "Phytophthora palmivora–Cocoa Interaction"

_jof, 2020, doi:10.3390/jof6030167_

Round 1

Reviewer 1 Report

I agreed to review this paper because of the title and the abstract stating that the paper would examine linkage between various infection structures and infection.  At best, there is a cursory review of spore germination and appressorium formation as well as general disease and host-pathogen interaction information.  The abstract also indicates the importance of the haustorium, but this is not well developed in the text. This is unfortunate because of the importance of these structures in both nutrient acquisition and in delivery of effectors.  the review does cover aspects of the morphology of the pathogen and different isolates that are causal agents of pod diseases, but also not in great detail. Even though there may be little information on this specific pathogen, the author could have developed a stronger paper by bringing in information from other Phytophthora species.

The review is a good start on the development of a review, but lacks depth and a critical analysis of the state of knowledge and what research questions need to be asked.

Author Response

I thank Reviewer 1’s for their comments and here are the changes made to the manuscript to improve this review.

Point 1: I agreed to review this paper because of the title and the abstract stating that the paper would examine linkage between various infection structures and infection.  At best, there is a cursory review of spore germination and appressorium formation as well as general disease and host-pathogen interaction information.  The abstract also indicates the importance of the haustorium, but this is not well developed in the text. This is unfortunate because of the importance of these structures in both nutrient acquisition and in delivery of effectors. 

Response 1:  The paper was modified by shortening the title and modifying the abstract on page 1, line 16 to 20. In addition, in the text a section was added to include the importance of the haustorium and the appressorium to deliver effectors and other proteins on pages 7-10 highlighted in red.  I also included aspects of PAMPs lines 140- 200 which are involved during infection. All references used are on page 18-20 – reference number [59] to [83].

Point 2: the review does cover aspects of the morphology of the pathogen and different isolates that are causal agents of pod diseases, but also not in great detail. Even though there may be little information on this specific pathogen, the author could have developed a stronger paper by bringing in information from other Phytophthora species.

Response 2: On pages 7-10, the whole section includes information from other Phytophthora species such as P. cinnamomi and P. infestans and the review links some of the structures, effectors and enzymes known to be released in other Phytophthora species to how P. palmivora could be infecting cocoa to cause disease. All references used are on page 18-20 – reference number [59] to [83].

Point 3: The review is a good start on the development of a review, but lacks depth and a critical analysis of the state of knowledge and what research questions need to be asked.

Response 3: By including reference [83] Ali and colleagues sequenced both Phytophthora megakarya and P. palmivora, which are closely related causal agents of cacao black pod rot, and were able to identify genes differentially expressed during infection in cocoa pod husks. I modified the final paragraph. This is where the research question is proposed that to understand the interaction of PNG Pal with cocoa in PNG, one would need to study the differential expression of genes in these isolates during infection in planta and their links to ETI and PTI.

Reviewer 2 Report

This is a nicely written review. I only have comments related to typos and grammars.

“Theobromae cacao” should be “Theobroma cacao”

Line 46, ‘Ppal sporangia typical shape range from ----“, need rephrase

Line 52, It may be better to remove “interestingly”. Reduction of high humidity is a common practice for Phytophthora disease control as high humidity/moisture and splashes of water help spread of zoospores. This is nothing out of the ordinary.

Line 56, Change “the sac which will contain…” to the present tense instead of using future tense.

Line 102, “Cyst germinate” should be “Cyst germinates”

Line 105, “ hyphae grows” should be “hyphae grow”

Line 105, “invade intracellular” should be “invade intracellularly”

Line 119-121, the sentence “Due to…” is unclear, need be rephrased.

Line 158, “…germinating zoospores, Ppal…” The comma should be “of”

Line 177, remove “that”, or rephrase the sentence.

Author Response

Response to Reviewer 2 Comments

This is a nicely written review. I only have comments related to typos and grammars.

Thank you for your comments and here are changes made to manuscripts (in red).

“Theobromae cacao” should be “Theobroma cacao” –

Spelling was checked and modified for “Theobroma cacao”.

Line 46, ‘Ppal sporangia typical shape range from ----“, need rephrase

The sentence was modified line 50-52 to ‘Sporangia morphology can be quite diverse but the shapes of Ppal sporangia range from ovoid-ellipsoid to obpyriform and they are papillate and cadacous i.e. short pedicels [1, 12, 13].

Line 52, It may be better to remove “interestingly”. Reduction of high humidity is a common practice for Phytophthora disease control as high humidity/moisture and splashes of water help spread of zoospores. This is nothing out of the ordinary.

Line 58, the word ‘interestingly’ was removed and the new sentence is, ‘Therefore, the reduction of high humidity and avoidance of excess water are some of the practices in greenhouses/glasshouses for Phytophthora disease control [19].’

Line 56, Change “the sac which will contain…” to the present tense instead of using future tense. -Line 63- change done

Line 102, “Cyst germinate” should be “Cyst germinates”- line 106, change done

Line 105, “ hyphae grows” should be “hyphae grow”- line 116, change done

Line 105, “invade intracellular” should be “invade intracellularly”, line 116, change done

Line 119-121, the sentence “Due to…” is unclear, need be rephrased.

Sentence was modified- line 133-135 to ‘There are differences in the structure and organisation of various plant tissues i.e. leaf, root, stem and pod so the focus will be on Ppal’s infection of a plant cell (Figure 2d-f).

Line 158, “…germinating zoospores, Ppal…” The comma should be “of” – line 257, done

Line 177, remove “that”, or rephrase the sentence.

Line 264-273 – that was removed and the sentences were rephrased to ‘Okey and colleagues [93] demonstrated a strong correlation between bark hardness and moisture content with canker resistance to Ppal in greenhouse studies. The same authors [ 93] proposed that extra-xylary tissue hardness associated with fibre content or deposition of suberin, callose and lignin could hinder the progress of fungal pathogens and that bark hardness had contributed to the slow rate of tissue colonization of Ppal in the canker resistant cocoa line used in the greenhouse studies leading to the use and selection of resistant cultivars with acceptable horticultural traits [94].’

Reviewer 3 Report

The paper synthetizes correctly the current information about the pathogen Phytophthora palmivora. However I would like the author to have in consideration some comments.

My main concern with the review is the section 4 (Overcoming plant host immunity by Ppal and other oomycetes). Section 4 contains a listing of molecular mechanism that take place during Oomycyte-plant interactions. This listing produces abrupt jumps that make the reading difficult. I would suggest introducing a sentence in line 221 to prepare the reader for the following list. It would be even more effective to produce a table similar to Table 1 in Raaymakers et al. 2016 (Extracellular Recognition of Oomycetes during Biotrophic Infection of Plants) where the components that will be discuss are presented (Nep1, lectins, elicitins, PsXEG1, GP42, RXLRs and CRNs). I would start with the general classification of RXLRs and CRNs (even though many do not fall into any of these categories) and then proceed to the more specific cases (Nep1, lectins, elicitins, PsXEG1, GP42). This table would also help clarifying when you are taking about plant proteins (i.e. lectin receptors in line 246) or pathogen proteins (i.e. Cellulose-binding lectins in line 252). The same problem exists with ELR and pep-13 (line 300 to 316).

The last paragraph of section 4 (line 359 to 376) should be added to the paragraph where PsXEG1 is discussed (line 285 to 299). It should be clarified that the enzymatic activity of PsXEG1 is antagonistic to its role as PAMP explaining why both silencing and overexpression of PsXEG1 produces reduction in virulence.

In line 417 it is mention that resistance to Ppal depends on lignin and peroxidase. However, nothing about the mechanism of defense based on induction of lignification is mention. Since this mechanism is explicitly mention to be important for Ppal, I would recommend mentioning this in section 4 citing papers like Egea et al. 2001 (Elicitation of peroxidase activity and lignin biosynthesis in pepper suspension cells by Phytophthora capsici).

Besides rewriting section 4 I would like to point out some minor modifications:

line 253 They have also been shown to aid in the Phytophthora’s penetration...

line 433 demonstrated that, when testing leaves and pods of various clones for resistance, there were two levels of resistance...

line 466 Phytophthora could gain allelic diversity and achieve large sexual/clonal population sizes...

line 488 In the case of characterized PNG Ppal...

Author Response

Response to Reviewer 3

The paper synthetizes correctly the current information about the pathogen Phytophthora palmivora. However I would like the author to have in consideration some comments.

Thank you for your comments and here are the changes.

My main concern with the review is the section 4 (Overcoming plant host immunity by Ppal and other oomycetes). Section 4 contains a listing of molecular mechanism that take place during Oomycyte-plant interactions. This listing produces abrupt jumps that make the reading difficult. I would suggest introducing a sentence in line 221 to prepare the reader for the following list. It would be even more effective to produce a table similar to Table 1 in Raaymakers et al. 2016 (Extracellular Recognition of Oomycetes during Biotrophic Infection of Plants) where the components that will be discuss are presented (Nep1, lectins, elicitins, PsXEG1, GP42, RXLRs and CRNs). I would start with the general classification of RXLRs and CRNs (even though many do not fall into any of these categories) and then proceed to the more specific cases (Nep1, lectins, elicitins, PsXEG1, GP42). This table would also help clarifying when you are taking about plant proteins (i.e. lectin receptors in line 246) or pathogen proteins (i.e. Cellulose-binding lectins in line 252). The same problem exists with ELR and pep-13 (line 300 to 316).

This section has been modified and Raaymakers et al 2016 reference added. Please refer to line 207-213.  Each topic - Nep1, lectins, elicitins, PsXEG1, GP42, RXLRs has a subheading from line 215 onwards.

The last paragraph of section 4 (line 359 to 376) should be added to the paragraph where PsXEG1 is discussed (line 285 to 299). It should be clarified that the enzymatic activity of PsXEG1 is antagonistic to its role as PAMP explaining why both silencing and overexpression of PsXEG1 produces reduction in virulence.

Paragraph is now under subheading d. GH12 proteins. Refer to line 329 onwards.

In line 417 it is mention that resistance to Ppal depends on lignin and peroxidase. However, nothing about the mechanism of defense based on induction of lignification is mention. Since this mechanism is explicitly mention to be important for Ppal, I would recommend mentioning this in section 4 citing papers like Egea et al. 2001 (Elicitation of peroxidase activity and lignin biosynthesis in pepper suspension cells by Phytophthora capsici).

This was added to section 4. “In another model system, it has been shown that peroxidase activity and lignin deposition increased in resistant Capsicum annuum (pepper) cell suspensions to P. capsici elicitors (Egea et al., 2001).” Please refer to lines 496-499.

Besides rewriting section 4 I would like to point out some minor modifications:

line 253 They have also been shown to aid in the Phytophthora’s penetration...

line 254- modified to “They have also been shown to be aid in Phytophthora’s penetration into its plant host by mediating the oomycete’s attachment to the host surface [115-117].”

line 433 demonstrated that, when testing leaves and pods of various clones for resistance, there were two levels of resistance...

line 527- modified to “Work by Iwaro and colleagues [144] tested leaves and pods of various clones for resistance and demonstrated that there were two levels of resistance in both organs.”

line 466 Phytophthora could gain allelic diversity and achieve large sexual/clonal population sizes...

Line 556 – modified to “Through sexual reproduction or interspecific hybridization, Phytophthora could gain allelic diversity and achieve large sexual/clonal population sizes through rapid proliferation [167]. These would enhance pathogen fitness by generating recombinant genotypes that may be more pathogenic or resistant to crop protection chemicals [168].”

line 488 In the case of characterized PNG Ppal...

line 610- modified to “In the case of the characterized PNG cocoa Ppal isolates [162], a study investigating the differences in gene expression related to PTI and ETI during cocoa pod and stem infections would be useful in understanding the differences in pathogenicity observed in cocoa plantation fields at different locations in PNG.”

Round 2

Reviewer 1 Report

This review is much improved in terms of the content, but still lack from a clear flow of ideas, a critical review of what is known, and how the information presented can be used to develop new lines of attack in the Phytophthora-Cocoa  interaction. These types of changes would result in a much better review.

I worked off of the version with track changes, and the line numbers refer to that version.

What follow are just a few comments, but the entire manuscript could use much more work as mentioned above.

The abstract makes no mention of the molecular and related studies now added to the text.

Lines 16-18 it is not clear why the author is suggesting a link between the germ tube, appressorium and haustorium as if these are novel to this system.  The pathogen could not set up a biotrophic phase without these elements.

19-20 the review spends relatively little time on these structures.

122-223 what is a “transient leaching treatment”?  Why would pectin be important since the pathogen is attaching to the cuticle?

124 what do you mean by “to dock”?

130  this implies that entry through wounds or stomata occur without appressoria – correct

133-135  the structures of these different tissues are potentially important in the development of disease.  Why only look at the host cell? And, which type or types of cell? Epidermis, mesophyll?

136 the appressorium peg would only be useful in penetrating the epidermis. What about penetration into ground tissues where haustoria are also found?

140-156  seems like a fairly superficial treatment of an important area of research related to recognition,

164-169  how do NLPs relate to your system?

170-173  does this also refer to penetration of epidermal cells where the pathogen first encounters a cuticle? Or are you referring to attachment in intercellular growth?

211-212 please provide some specific examples of these effectors during the formation of appressoria and haustoria development. The rest of the paragraph could be more in depth in terms of the significance  of the work

265  are you implying that bark is a physical barrier? Are you referring to the outer rhytidome or the entire periderm plus cambial tissues and phloem?

274  How does the work on phosphite follow from the first part of this paragraph,  and how does the rest of the paragraph  follow from this statement on phosphite. 

Author Response

RESPONSE TO REVIEWER 1’S REPORT

This review is much improved in terms of the content, but still lack from a clear flow of ideas, a critical review of what is known, and how the information presented can be used to develop new lines of attack in the Phytophthora-Cocoa  interaction. These types of changes would result in a much better review.

I worked off of the version with track changes, and the line numbers refer to that version.

What follow are just a few comments, but the entire manuscript could use much more work as mentioned above.

Thank you for the suggestions and these are the changes applied to the revised version of the manuscript. Responses are in red.

The abstract makes no mention of the molecular and related studies now added to the text.

The abstract has been modified to reflect molecular and related studies included in the text. Please refer to line 14- 22.

Lines 16-18 it is not clear why the author is suggesting a link between the germ tube, appressorium and haustorium as if these are novel to this system.  The pathogen could not set up a biotrophic phase without these elements.

The sentence was reworded to highlight the importance of the germ tube, the appressorium and the haustorium in establishing disease. Please refer lines 14- 22.

19-20 the review spends relatively little time on these structures.

Two extra keywords were added. Stem canker as this including Black pod rot are the symptoms of disease when Phytophthora palmivora infects cocoa and the word infection – line 22-23.

122-223 what is a “transient leaching treatment”? 

An explanation for transient leaching treatment was added to text. Please refer to line 122- 124.

Why would pectin be important since the pathogen is attaching to the cuticle?

An explanation about pectin, its presence in the cuticle and Phytophthora’s ability to degrade pectin such as pectin methylesterase has been included in the text. Please refer to line 126 -132 as well as references 49-53.

124 what do you mean by “to dock”?

An explanation about ‘to dock’ was included in the text in relation to precise orientation of zoospores on the plant surface. Please refer to line 118-121 and references 42-44.

130  this implies that entry through wounds or stomata occur without appressoria – correct

Extra information was added regarding entry of Phytophthora via stomata and wounds as germ tubes gain entry in the space. Please refer to line 137-141 and references 58 and 59.

133-135  the structures of these different tissues are potentially important in the development of disease.  Why only look at the host cell? And, which type or types of cell? Epidermis, mesophyll?

This section was modified to providing more details in the infection process and disease development in various tissue. Please refer to lines 141-157 and references 60 to 72. All living plant cells can be invaded by Phytophthora and the Fig 2. gives a general process of infection and the formation of the haustorium in plant cells.

136 the appressorium peg would only be useful in penetrating the epidermis. What about penetration into ground tissues where haustoria are also found?

This was also address in the above section – lines 141 -157 and references 60-72. Furthermore, the legend of Figure 2 was modified to mention about ground tissues. Please refer to lines 110 to 112 ‘invasion into the epidermis of some aerial tissues. In ground tissues, the germ tube can penetrate the root epidermis by growing intercellularly along the anticlinal cell walls’ . Extra references were added to the paper (refer to 73, 75 and 83) and the section expanded to include details mentioned above. Please refer to line 132- 162.

140-156  seems like a fairly superficial treatment of an important area of research related to recognition,

Extra references were added and the section expanded to include some aspects of NB-LRR and MiRNAs, the link between SAR and HR/ immunity and infection of Medicago by rhizobia. Please refer to line 163-200 and references 84-97.

164-169  how do NLPs relate to your system?

Two extra references were added and the role of NEP1 in cocoa work mentioned in this section. Please refer to lines 213-223 and references 104 and 105.

170-173  does this also refer to penetration of epidermal cells where the pathogen first encounters a cuticle? Or are you referring to attachment in intercellular growth?

According to reference 75, in Fig. 2 PsXLP1 and PsXEG1 transcript levels were low in mycelium and zoospores but transcripts levels increased after 10 mins post cyst germination and decreased after 6 h and only PsXLP1 transcripts remained elevated up to 48 hours. This would relate to contact/penetration after germination of the zoospores and growth up to 48 h.

211-212 please provide some specific examples of these effectors during the formation of appressoria and haustoria development. The rest of the paragraph could be more in depth in terms of the significance of the work

More information and examples were included between lines 224 -266 on plant lectins, CBELs and links to BAK1 as well as pep-13. Extra references – reviews which investigated effectors and appressoria and haustoria development were added in context to infection and how the haustorium plays a critical role in successful parasitic infection by Phytophthora – see lines 157- 162.

Extra section was added highlighting some work by Ochola et al 2020 – refer to lines 302-319 and reference 132.

265  are you implying that bark is a physical barrier? Are you referring to the outer rhytidome or the entire periderm plus cambial tissues and phloem?

More information was added to this section – providing some detail about stem canker and how the cambial layer is infected. Traits selected were based lignin and the enzyme activities related to lignin and phenol. Please refer to lines 329, 332-334 and 337 -343 as well as references 141-146.

274  How does the work on phosphite follow from the first part of this paragraph,  and how does the rest of the paragraph  follow from this statement on phosphite.

This was modified – please refer to line 345 to 348 in text and how it is used to control stem canker.

Reviewer 2 Report

Compared with the initially submitted version, the author added multiple paragraphs under “P. pal and its infection process in cocoa.” The added parts were poorly written and much of them may not be relevant to P. pal at all. The fact that a gene was identified in another oomycete does not suggest that a homolog is for sure present in P. pal. As the genomes of P. pal from cacao isolates were published (Publication 1: Genome Biol Evol. 2017 Feb 10;9(3):536-57. Publication 2: G3 (Bethesda) 2020 Jul 7;10(7):2241-2255.), the author should look closely at those data to see if the homologs of a particular gene to be included in the manuscript actually exist in P. pal and whether the expression pattern suggests the involvement of P. pal-cocoa interactions.

Specific comments are below:

In line 150-154, the author only mentioned the homologs of NLPs in other plant species, not even mentioned the NLPs in P. pal. The P. pal homologs were described in Publication 1 (Above).

Line 155-158, CBELs may not be conserved in Phytophthora spp. Unless the author identified them in the genomes of P. pal, this part need be deleted. Similar concern applies to Line 166-172, Line 173-177, and 203-206.

In Line 159-165, the author discussed about elicitin, a PAMP. In line 178, “palmivorein” was mentioned. Pamivorein is an elicitin of P. pal. These need be combined.

In Line 193-205, the author seemed to switch from PAMPs to effectors (RXLR and CRN effectors). However, elicitin was mentioned again (Line 200). Line 193-194, “Many effectors are known to act in the apoplastic and symplastic region of plant cells during the appressorium and the haustorium development.” The author should provide a reference. Based on the knowledge I have, the statement is incorrect.

Line 197, “CRickling”, should be “CRinkling”.

Line 200, “putative extracellular proteins”, change “extracellular” to “secreted”. The RXLR and CRN effectors are secreted from the pathogen, but act inside the host cells, therefore, they are not extracellular.

The author need read the above publications (and check other relevant references) closely and analyze the genome data when necessary to be able to extract the knowledge and present in an accurate and well-organized manner.

Author Response

RESPONSE TO REVIEWER’S COMMENTS

Compared with the initially submitted version, the author added multiple paragraphs under “P. pal and its infection process in cocoa.” The added parts were poorly written and much of them may not be relevant to P. pal at all. The fact that a gene was identified in another oomycete does not suggest that a homolog is for sure present in P. pal. As the genomes of P. pal from cacao isolates were published (Publication 1: Genome Biol Evol. 2017 Feb 10;9(3):536-57. Publication 2: G3 (Bethesda) 2020 Jul 7;10(7):2241-2255.), the author should look closely at those data to see if the homologs of a particular gene to be included in the manuscript actually exist in P. pal and whether the expression pattern suggests the involvement of P. pal-cocoa interactions.

Information from Publication 1 and publication 2 was added to the review and references included.

Specific comments are below:

In line 150-154, the author only mentioned the homologs of NLPs in other plant species, not even mentioned the NLPs in P. pal. The P. pal homologs were described in Publication 1 (Above).

Publication 1 and information about NLPs in P. pal were added. See lines 436 -442

Line 155-158, CBELs may not be conserved in Phytophthora spp. Unless the author identified them in the genomes of P. pal, this part need be deleted. Similar concern applies to Line 166-172, Line 173-177, and 203-206.

These concerns about CBELs and various other proteins were addressed in lines 452- 455, lines 484 and 743-748, lines 763-765.

In Line 159-165, the author discussed about elicitin, a PAMP. In line 178, “palmivorein” was mentioned. Pamivorein is an elicitin of P. pal. These need be combined.

Regarding elicitin and palmivorein, these were combined – please refer to line 459-476.

In Line 193-205, the author seemed to switch from PAMPs to effectors (RXLR and CRN effectors). However, elicitin was mentioned again (Line 200).

The word elicitin was removed.

 Line 193-194, “Many effectors are known to act in the apoplastic and symplastic region of plant cells during the appressorium and the haustorium development.” The author should provide a reference. Based on the knowledge I have, the statement is incorrect.

Two references were added – please refer to line 717 – reference numbers [61, 86].

Kebdani N, Pieuchot L, Deleury E, Panabières F, Le Berre JY, Gourgues M (2010) Cellular and molecular characterization of Phytophthora parasitica appressorium-mediated penetration. New Phytol 185: 248–257

Wang, S., Boevink, P.C., Welsh, L., Zhang, R., Whisson, S.C. and Birch, P.R.J. (2017), Delivery of cytoplasmic and apoplastic effectors from Phytophthora infestans haustoria by distinct secretion pathways. New Phytol, 216: 205-215. doi:10.1111/nph.14696

Line 197, “CRickling”, should be “CRinkling”.

The word has been changed to “CRinkling” – please refer to line 720.

Line 200, “putative extracellular proteins”, change “extracellular” to “secreted”. The RXLR and CRN effectors are secreted from the pathogen, but act inside the host cells, therefore, they are not extracellular.

The words have been change to “putative secreted proteins”. Please refer to line 723.

The author need read the above publications (and check other relevant references) closely and analyze the genome data when necessary to be able to extract the knowledge and present in an accurate and well-organized manner.

Publication 1 database was checked and key information added as mentioned above. And database from Evangelisti et al 2017 was also searched and information added.

Extra references for paper

Kelley, B.S.; Lee, S.‐J.; Damasceno, C.M.B.; Chakravarthy, S.; Kim, B.‐D.; Martin, G.B.; Rose, J.K.C. A secreted effector protein (SNE1) from Phytophthora infestans is a broadly acting suppressor of programmed cell death. Plant J 2010, 62: 357-366. doi:10.1111/j.1365-313X.2010.04160.x

Zerillo, M. M., Adhikari, B. N., Hamilton, J. P., Buell, C. R., Lévesque, C. A., & Tisserat, N. (2013). Carbohydrate-active enzymes in pythium and their role in plant cell wall and storage polysaccharide degradation. PloS one8(9), e72572. https://doi.org/10.1371/journal.pone.0072572

Khatib, M., Lafitte, C., Esquerré-Tugayé, M.-T., Bottin, A., and Rickauer, M. (2004). The CBEL elicitor of Phytophthora parasitica var. nicotianae activates defence in Arabidopsis thaliana via three different signalling pathways. New Phytol. 162 501–510.

Morales-Cruz, A., Ali, S., Minio, A., Figueroa-Balderas, R., García, J., Kasuga, T., Puig, A., Marelli, J., Bailey, B., & Cantu, D. (2020). Independent Whole-Genome Duplications Define the Architecture of the Genomes of the Devastating West African Cacao Black Pod Pathogen and Its Close Relative. G3 (Bethesda, Md.)10(7), 2241–2255. https://doi.org/10.1534/g3.120.401014

Round 3

Reviewer 2 Report

The revised manuscript improved greatly compared to the previous versions.

Author Response

The revised manuscript improved greatly compared to the previous versions.

Thank you for your helpful comments